# Use of Biostimulants as a New Approach for the Improvement of Phytoremediation Performance—A Review

**DOI:** 10.3390/plants11151946

**Published:** 2022-07-27

**Authors:** Maria Luce Bartucca, Martina Cerri, Daniele Del Buono, Cinzia Forni

**Affiliations:** 1Department of Agricultural, Food and Environmental Sciences, University of Perugia, Borgo XX Giugno 74, 06121 Perugia, Italy; marialucebartucca@gmail.com (M.L.B.); cerrimartina@gmail.com (M.C.); daniele.delbuono@unipg.it (D.D.B.); 2Department of Biology, University of Rome Tor Vergata, Via della Ricerca Scientifica, 00133 Rome, Italy

**Keywords:** biostimulants, phytoremediation, pollutants, plant stress

## Abstract

Environmental pollution is one of the most pressing global issues, and it requires priority attention. Environmental remediation techniques have been developed over the years and can be applied to polluted sites, but they can have limited effectiveness and high energy consumption and costs. Bioremediation techniques, on the other hand, represent a promising alternative. Among them, phytoremediation is attracting particular attention, a green methodology that relies on the use of plant species to remediate contaminated sites or prevent the dispersion of xenobiotics into the environment. In this review, after a brief introduction focused on pollution and phytoremediation, the use of plant biostimulants (PBs) in the improvement of the remediation effectiveness is proposed. PBs are substances widely used in agriculture to raise crop production and resistance to various types of stress. Recent studies have also documented their ability to counteract the deleterious effects of pollutants on plants, thus increasing the phytoremediation efficiency of some species. The works published to date, reviewed and discussed in the present work, reveal promising prospects in the remediation of polluted environments, especially for heavy metals, when PBs derived from humic substances, protein and amino acid hydrolysate, inorganic salts, microbes, seaweed, plant extracts, and fungi are employed.

## 1. Introduction

### 1.1. Environmental Degradation and Pollution

In recent decades, we have witnessed a significant increase in world population and economic growth [1]. There is a close relationship between economic growth, energy consumption and environmental degradation, as highlighted by some recent studies [2,3,4,5]. This interdependence has become a significant public policy priority among Organisation for Economic Co-operation and Development (OECD) countries [4], since environmental degradation is one of the most significant challenges that humans have to face in the near future [5]. The main evidence of environmental degradation is the depletion and pollution of natural resources, destruction or degradation of ecosystems, and extinction of wildlife [6]. The risks associated with environmental pollution include contamination of food, air, water, and soil, raising the question of identifying effective strategies to prevent and mitigate these problems [7].

Heavy metals (HMs) are globally considered among the most relevant pollutants [8,9]. Human activities such as fossil fuel combustion, mining and smelting of metal ores, urban and industrial expansion, the production of large amounts of municipal waste, and agricultural practices constantly release HMs into the environment [10,11]. Currently, HMs are considered the main pollutants in European soils and groundwater [12]. HMs can be toxic to living organisms at very low concentrations, depending on their chemical and physical properties [13]. In plants, they can lead to oxidative stress, hinder plant growth, interfere with photosynthetic activity, replace other metals in pigments or enzymes, and accelerate senescence [14].

In addition to HMs, pesticides can be detected in water and soil in concentrations that often exceed the limits admitted by law [15,16]. The accumulation of pesticides in soil and water should be avoided to prevent them from entering the food chain, which can cause, in turn, severe threats to human and animal health [17]. Although pesticides are applied in agriculture to control and limit weed competition with crops, their selectivity in targeting weeds is in some cases only theoretical. In fact, certain pesticides can reach non-target organisms, such as crops, thus provoking morphological, physiological, and biological alterations [18]. For instance, it has been shown that pesticides can reduce crop growth and biomass production, interfere with mineral nutrition and use efficiency, reduce chlorophyll concentration, and cause cell death [19,20].

The implementation of more stringent legislation is necessary to reduce polluting emissions into the environment [21]. The switch to renewable energy sources is another critical element in reducing environmental pollution [22]. Modeling point and non-point sources of pollution is essential to assess their effect on air, soil, and water ecosystems and make decisions about the measures and actions to be taken [23]. In addition to reducing emissions, particular attention must be paid to the remediation of polluted environments [24]. Many methods can be currently applied to clean or recover polluted environments, and they include ex situ and in situ remediation techniques: “dig and dump”, “pump and treat”, chemical oxidation/reaction, incineration, thermal treatment, dilution or chemical stabilization/immobilization of the contaminant, electrokinetics, soil washing/flushing, excavation, and disposal [25,26]. Nonetheless, these methods can be expensive, consume energy, require specific machinery, and may negatively impact the environment or cause secondary pollution [27,28]. For instance, when applied to polluted soil, most of the abovementioned physical or chemical techniques affect its biological activity, structure, and fertility [27,28,29]. Furthermore, the removal of the contaminant may not be entirely satisfactory [30].

### 1.2. Phytoremediation

In recent years, concerns related to ecological threats have led to searching for new and cheap bio-based remediation technologies [29]. In this context, *bioremediation*, which refers to the use of microbes to clean polluted environments, has gained increasing attention and diffusion [31]. In fact, some bacteria, fungi, archaea, and algae may show a high capacity to remove or neutralize many types of contaminants [32]. In *bioremediation*, the cleaning process is mainly due to the action of specific enzymes naturally occurring in microorganisms [33].

Another green technique suitable for the recovery of polluted sites is *phytoremediation*. This emerging and biological-based technology exploits the ability of plants to decontaminate air, soil, and water of various kinds of contaminants or transform them into less toxic compounds or derivatives [34]. *Phytoremediation* is highly appreciated for its effectiveness and eco-friendliness [35]. Indeed, it has been demonstrated that certain plant species can remove, reduce, or stabilize significant amounts of contaminants from polluted sites. Species suitable for phytoremediation programs must cope with the adverse effects caused by toxicants, which could otherwise seriously hamper their vital functions [36]. Usually, these species constitutionally express high amounts of antioxidant or detoxifying molecules and enzymes. The presence of the xenobiotics can also induce the cellular content of these protective activities as a defensive response [37].

Several different phytoremediation techniques are applied to remove a wide range of pollutants or transform them into less harmful derivatives [21]. An in-depth study and exploitation of the processes involved in phytoremediation is not the purpose of this review; we refer for further information and details to the vast published literature [21,38,39,40]. However, the main phytoremediation techniques are briefly summarized below.

*Phytoextraction*: this technology is based on the capacity of some plants to absorb contaminants by roots from polluted sites and translocate them to the aboveground tissues. This ability is generally suitable for remediating sites polluted by heavy metals. Moreover, this approach offers other interesting opportunities: it allows the recovery of metals in saline form through plant harvesting and acid digestion of plant tissues. This technique is called *phytomining* and has recently gained a great interest if the metals are precious or of technological value [41,42]. The so-called *hyperaccumulators* are plants particularly appreciated in phytoremediation. These species are generally employed in phytoextraction since they can remove and store in their tissues very high amounts of toxic substances [43]. In the use of hyperaccumulators, additional strategies can be applied to improve the phytoremediation efficiency. For example, it is possible to modulate the plant uptake of contaminants by acting on fertilization [44]. This is the case for arsenic (As) absorption, which depends on the amounts of phosphorous (P) available to the plant [44]. In fact, As is analogue to P and it is taken up by the plant through P transporters [45]. Thus, the two elements compete for the same carriers in roots [46] and by reducing the phosphorus supply to the plant it is possible to increase As remediation [45].

Plants with high tolerance to contaminants, efficient translocation from root to shoot, effective detoxification, and large biomass production can efficiently bioconcentrate the target substance. A widely employed parameter accounting for this ability is the bioconcentration factor (BCF) [28]. The BCF represents the ratio between the concentration of a target substance in plant tissues and that in the growth medium [47]. In phytoremediation, this parameter is considered a useful index to assess the capacity of a plant to remove a pollutant [48]. A metal hyperaccumulator, for example, can accumulate in its tissues more than 10.000 mg kg^−1^ of manganese (Mn) or zinc (Zn) without showing significant alterations in the metabolic activity or physiological functions [49].

*Phytostabilization*: this technology is based on the use of plants that can absorb or precipitate the pollutant, immobilizing it in the rhizosphere [21,50]. This action strongly decreases the bioavailability of the toxic substance, thus preventing the contaminant from reaching the groundwater or entering the food chain [27].

*Phytodegradation*: this technique exploits the ability of plants to metabolize or detoxify the xenobiotic thanks to the action of various enzymes (dehalogenase, peroxidase, glutathione *S*-transferase, etc.). Phytodegradation is usually applied to remove organic pollutants, such as herbicides, from polluted environments, thanks to the ability of certain plants to inactivate these substances and sequester/immobilize them [43].

*Phytovolatilization*: this technology is based on the ability shown by certain species to take up the pollutants from the growth media by roots and transform them into volatile forms. Then, the xenobiotics can be released into the atmosphere by the stomata [42]. This technique is frequently applied to remove some metals and metalloids (mercury, Hg, selenium, Se, or arsenic, As) or volatile organic compounds (VOCs) from polluted sites [27].

*Rhizofiltration*: this technology regards the use of aquatic and terrestrial plants to clean aqueous media of contaminants. Alternatively, the plant may precipitate the target substance, limiting its mobility and bioavailability [43]. This method can be particularly effective in removing heavy metals, dyes, and organic compounds [27,51].

*Phytostimulation* (or *rhizodegradation*): this technology regards the degradation of the pollutants by bacteria and fungi living in the rhizosphere [52]. These microorganisms take advantage of the substances naturally exuded by plant roots (such as sugar or amino acids) that serve them as nourishment and enhance their metabolism and biological activity [27]. Phytostimulation is generally employed for the remediation of soils polluted by organic compounds such as pesticides, polycyclic aromatic hydrocarbon (PAH), or polychlorinated biphenyls (PCBs) [43].

Despite the considerable knowledge acquired to date, many processes involved in the different phytoremediation methods still remain to be fully elucidated [21]. If compared to conventional methods, phytoremediation shows some significant advantages and disadvantages, which are reported in Table 1 [21,27,29,38,53,54].

### 1.3. Emerging Tools to Improve Plant Efficiency in Phytoremediation Programs

In recent years, many studies have paid attention to reducing or compensating for the disadvantages and limitations characterizing phytoremediation. Consequently, new approaches have been developed and applied to increase the effectiveness of this technology or broaden its application horizon [42]. For instance, among these approaches, the herbicide-safeners have been recently proposed and successfully tested [8,21,24,55]. These synthetic compounds are commercial products specifically applied to cereal crops to improve their tolerance to herbicides routinely employed in weed control [56]. It has been shown that these chemicals can enhance herbicide metabolism in plants thanks to the specific induction of enzymes involved in xenobiotic detoxification. Moreover, safeners may help plant species cope with oxidative stress caused by xenobiotics: studies have shed light on the ability of these compounds to stimulate antioxidant defenses [24,56]. Consequently, plants can increase pollutant uptake thanks to their enhanced resistance to toxic substances. This beneficial effect results in an improved phytoremediating performance [21]. The advantage of using safeners is that they have been classified as “environmentally inert”; furthermore, they are quickly degraded or transformed into compounds that do not affect the ecosystem and the health of living organisms [57].

Genetic improvement through conventional plant breeding is another way to enhance the performance of plants in phytoremediation; however, in recent years, genetic engineering techniques have also been proposed. These two methods can even be successfully combined in phytoremediation [58]. Genetic engineering implies using transgenic plants, which are species genetically modified by recombinant DNA technologies. In the plant selected for phytoremediation, genes of other organisms (plants or bacteria) or endogenously modified, mainly involved in the acquisition, translocation, detoxification, and concentration of the contaminants, are transferred or overexpressed [28,59]. This method allows for overcoming some plant limits in phytoremediation [60]. For instance, in transgenic plants, genes encoding for enzymes involved in the synthesis of metallothioneins (MTs), phytochelatins (PCs) (both active in metal chelation), or glutathione (GSH) (for its crucial role in antioxidative and detoxification mechanisms) can be overexpressed. This modification improves the species’ resistance to contaminants and efficiently enhances their capacity to remove pollutants from contaminated sites [28]. In this context, the use of transgenic hyperaccumulators has been proposed (*genoremediation*) [61]. This approach presents significant advantages (especially in cleaning environments polluted by heavy metals); however, it may entail high direct and indirect risks, which are discussed in more detail in the specific literature [28]. In this regard, the omic approach (genomic, transcriptomic, proteomic, and metabolomic), combined with bioinformatics tools, is advantageous since it could allow a deeper understanding of the metabolism of both plants and soil microorganisms. Such an aspect allows analyzing the possibilities to maximize the co-operative potential for bioremediation purposes [62].

In recent years, some studies have proposed biostimulants as interesting and novel candidates to enhance the phytoremediation efficiency of polluted environments. In particular, previous literature has shown their effectiveness in increasing plant vigor and tolerance to various abiotic and biotic stresses [63,64,65,66]. This property can be exploited to increase the resistance of biostimulated species to toxic substances. In addition, such an intriguing property can allow planning to use biostimulants to increase the capacity of the species used in phytoremediation to remove toxic substances from polluted sites.

However, to our knowledge, no reviews have been published to date on this subject. Therefore, in the present work, the possibility of exploiting plant biostimulants as a new tool to increase plant resistance to toxic substances and improve their performance in phytoremediation is reviewed and discussed. To this end, after a brief introduction aimed at explaining what biostimulants are, the scientific studies that have tested their effectiveness in phytoremediation to date will be reported and discussed.

## 2. Plant Biostimulants

Plant biostimulants (PBs, also referred to as biofertilizers, biostimulators, plant probiotics, or metabolic enhancers [67]) are materials currently employed in agriculture with the scope of improving plant productivity and quality [68]. In fact, their application to crops makes it possible to trigger physiological and molecular processes that can positively affect yield and product quality [69]. The recent EU Regulation (2019/1009) defined PBs as “*fertilising product the function of which is to stimulate plant nutrition processes independently of the product’s nutrient content with the sole aim of improving one or more of the following characteristics of the plant and/or the plant rhizosphere: (1) nutrient use efficiency, (2) tolerance resistance to (a) biotic stress, (3) quality characteristics, or (4) availability of confined nutrients in the soil or rhizosphere*”. PBs do not fall into the category of fertilizers or plant protection products, as their primary function is neither to provide nutrients nor to protect plants from pests and pathogens [69]. These products contain substances and/or microorganisms that enhance nutrient availability to plant roots (and consequently their uptake), stimulate the plant’s capacity to use nutrients, and, in some cases, to cope with abiotic stresses [70]. Furthermore, PBs, for their ability to efficiently improve plant nutrient acquisition, could permit reductions in the use of chemical fertilizers routinely employed in agriculture. Consequently, this could also promote the environmental sustainability of agriculture, thanks to the possible reduction in synthetic compounds consumed in large amounts by this activity [71]. Although PBs were initially adopted in horticulture, to date, these substances have been used to stimulate beneficial effects in a wide range of crops. The PB global market was estimated at USD 2.6 billion in 2019 [67].

The substances that can exert stimulatory effects on plants can be obtained starting from raw materials of profoundly diverse origins and compositions [69]. For this reason, biostimulants have been grouped into different families: humic and fulvic substances, inorganic salts, protein hydrolysates and amino acids, complex organic materials, seaweed and plant extracts, chitin and chitosan derivatives, organic acids, animal/vegetal protein, and beneficial microorganisms (bacteria such as *Bacillus* and *Azotobacter* spp., yeast, filamentous fungi, and micro-algae) [72,73].

The beneficial effect of biostimulants on plant growth, productivity, yield, and quality may also depend on the synergistic actions of their multiple constituents. Consequently, in general, the mode of action of biostimulants is still unknown [74]. For this reason, a biostimulant is regarded as such only for its beneficial effect on crops, when demonstrated, which is to improve plant nutrient acquisition, production, and resistance to adverse environmental conditions [75].

### PBs in Helping Crops Cope with Toxic Compounds

In this section, particular attention is paid to the ability of PBs to increase plant resistance to various kinds of stress. Biostimulants, applied in small amounts to plants, seeds, or the rhizosphere, can stimulate the crop’s tolerance to adverse environmental conditions, such as salinity, drought, extreme temperatures, and UV radiation [68,73,75]. For example, salt and drought stress can affect key physiological and biochemical processes in plants, such as chlorophyll and pigment biosynthesis, leaf gas exchange, relative water content, or antioxidant enzymes’ activity and determine water loss and lipid membrane oxidation. Recent studies have reported that PBs could alleviate these damages [76,77]. Furthermore, PBs could allow the reduction in the use of chemical fertilizers, due to the improved efficiency of biostimulated crops in acquiring and using nutrients [68].

PBs have also been tested on plants directly grown in polluted environments, thus exploiting their capacity to help species cope with toxic substances. Some of these studies were conducted in soils/water contaminated by heavy metals (HMs) or pesticides which, as mentioned in the Introduction, are among the main world pollutants.

Recent studies indicated that PBs could reduce heavy metal (HM) toxicity to plants. Calvo et al. [70] documented that this beneficial effect could be prompted by protein-based products and fulvic and humic acids. In addition, biostimulants containing peptides and amino acids as active ingredients can enhance plant tolerance to HMs. Among the amino acids, the increase in the amount of proline in plants is particularly effective in their protection since it can chelate metal ions within plant cells, function as an antioxidant, and play a pivotal role in osmoregulatory processes [70]. Moreover, plant biostimulants based on humic substances (HSs, which include humic acids (HAs) and fulvic acids (FAs)) can induce the activity of specific antioxidant enzymes (peroxidase, catalase) and increase the content of non-enzymatic antioxidants, which are essential for plant survival. In particular, the increase in the content of non-enzymatic antioxidants results from the ability of HSs to stimulate the synthesis of compounds linked to the shikimic acid pathway (alkaloids, phenols, and tocopherols) [78]. HSs can interact with HMs and complex them with carboxylic and phenolic hydroxyl groups, and this can decrease the mobility of HMs in soil and, consequently, their bioavailability to plant roots [78].

Canellas et al. [79] proposed the maize (*Zea mays*) “priming”, which consists of treating the seeds with appropriate doses of HSs. The priming, for its effectiveness, is considered a functional strategy to improve crop performance and protect them from the detrimental effects of various abiotic stressors [79]. In this study, HA-primed plants showed increases in the antioxidant enzyme catalase (CAT) and proline. Biostimulated plants also showed higher transcription levels of genes associated with stress signaling and response. In particular, the authors found increases in the expression of genes encoding for proteins involved in autophagy processes and for kinase, phosphatase, and phytohormones (auxin, abscisic acid, ethylene). This effect occurred even when the plants were biostimulated in the absence of stressors. Finally, HA-primed plants exposed to various stresses, including HMs, showed improved resistance and higher biomass production [79].

Arbuscular mycorrhizal fungi (AMF) can play an essential role in protecting plants from the injuries caused by HMs [70]. AMF can exert beneficial effects in plants by immobilizing metals, thus reducing their availability to roots [80]. In addition, biostimulants based on plant growth-promoting rhizobacteria (PGPRs) can reduce HM absorption by roots and their translocation in the aboveground tissues since they can chelate, bind, and precipitate them [81]. Consequently, the beneficial action of PGPRs is also reflected by lower amounts of HMs in the aboveground tissues of biostimulated plants [81].

PBs derived from plant extracts have been found to be able to protect plants from HM toxicity. In a very recent study, a silymarin-based biostimulant was found to attenuate the damages caused by cadmium (Cd) to maize plants [82]. PB application effectively restored Cd-exposed plants by stimulating biomass production, hormone homeostasis, photosynthetic efficiency, and inducing the activity of certain antioxidant enzymes.

Other studies indicated that PBs can increase crop tolerance to pesticides [21]. Likewise, a recent study has demonstrated that a commercial biostimulant (Megafol) improved maize resistance to the chloroacetanilide herbicide metolachlor [83]. In biostimulated plants, the authors found lower levels of lipid membrane peroxidation and increased germination, biomass production, and vigor index with respect to maize treated with the herbicide alone [83]. This beneficial effect was attributed to the induction of some antioxidant enzymes (ascorbate peroxidase—APX, guaiacol peroxidase—GPX, and catalase—CAT) found in the biostimulated plants [83]. Another study showed that the biostimulant Fertiactyl Pós^®^ reduced the injuries caused by the herbicide glyphosate to soybean (*Glycine max*) [84]. This PB contains humic and fulvic acids, which can bind the herbicide, and glycine, betaine, and zeatin, which help plants to overcome oxidative stress [85]. In the cited study, Fertiactyl Pós^®^ prevented yield losses and limited the symptoms of chlorosis and necrosis resulting from the herbicide application [84]. In fact, glyphosate can directly damage the chlorophyll or reduce the availability of nutrients involved in its functioning (Mg and Mn) [84]. Balabanova et al. [86] reported that a PB based on amino acids protected sunflower (*Helianthus annuus*) from the damages caused by the herbicide imazamox. The biostimulant exerted its beneficial action by restoring the net photosynthetic rate, stomatal conductance, chlorophyll content, and plant growth. The authors stated that further studies should be carried out in order to understand and clarify the mechanism of action of the investigated product [86]. The studies published to date on the effectiveness of PBs in reducing the stress generated by pollutants in plants are summarized in Table 2.

In addition to the findings mentioned above, which document the efficacy of PBs in increasing plant resistance to pesticides, these materials can reduce the toxicity of these xenobiotics to plants by acting directly on the soil. In a study conducted by Tejada et al. [87], the herbicide MCPA depressed soil enzymatic activities and the ergosterol content. The application of four biostimulants of different origin and composition (wheat condensed distillers soluble, WCDS; hydrolyzed poultry feathers, PA-HE; carob germ enzymatic extract, CGHE; and rice bran extract, RB) counteracted these adverse effects. The most effective biostimulant (PA-HE) showed a higher protective action thanks to its higher content of low molecular weight peptides, humic substances, and lower fat content [87]. In another study, Rodriguez-Morgado et al. [88] observed that two biostimulants (SS, derived from sewage sludge; and CF, derived from chicken feathers) mitigated the negative impact of the herbicide oxyfluorfen on soil enzymatic activities and microbial communities. In a review article, Kanissery and Sims [89] pointed out that soils showed a higher rate of herbicide removal in cultivated crop fields when plants were treated with biostimulants. This effect was attributed to the ability of the organic material to provide nutrients to microorganisms that populated the soil, thus increasing their degradation activity towards the herbicide. Based on their results, the authors concluded that biostimulants could be seen as a promising and effective tool to promote soil cleaning of herbicides [89].

Due to the suitability of biostimulants in reducing the toxicity of the pollutants to plants, it can be advantageous to explore the use of these materials to improve the plant performance in the remediation of contaminated sites. Based on this, further investigation needs to be considered in phytoremediation programs. In this scope, in the following sections, scientific studies on the PBs’ applicability to potentiate this technique will be reported and discussed.

## 3. PBs for Phytoremediation

In the previous section, the effects of PBs in helping plants counteract various types of stress, including those generated by the presence of pollutants, have been described. As a consequence, their activity may ameliorate phytoremediation performance as well (Figure 1).

### 3.1. PBs Derived from Humic Substances

Humic substances are complex biomolecules derived from the biological and chemical degradation of plant and animal residues in soil [90]. Further than being the main components of soil organic matter, they represent the major carbon pool in the biosphere [75]. HSs consist of mixtures of (a) humic acids (HAs), (b) fulvic acids (FAs), and (c) humin. HAs are typically high molecular weight (10,000–100,000 g/mol) compounds, soluble in alkaline solutions; FAs have low molecular weights (1000–10,000 g/mol), and are soluble in all pH conditions, while humin is insoluble due to its very high molecular weight (100,000–10,000,000 g/mol) [91,92]. HSs play many fundamental roles in soil, regulating, for instance, the nitrogen and carbon cycle and the oxygen exchange with the atmosphere. Furthermore, they support microbial communities, stimulating their growth and affecting their primary and secondary metabolism. HSs promote many beneficial effects in plants, such as favoring seed germination, enhancing nutrient uptake, supporting plant growth and development, improving product quality and yield, and ameliorating stress resistance [70,71,72,73,74,75,76,77,78,79,80,81,82,83,84,85,86,87,88,89,90]. HSs can also influence the fate of toxic substances by regulating their transport and stabilization in soil and, consequently, their effect on plants and soil-populating bacteria [78]. Pittarello et al. [93] exploited this behavior towards mangrove sediments and found that the sediments’ absorbing capacity of copper (Cu), cadmium (Cd), and lead (Pb) was significantly increased by the addition of HSs. In another study, the same authors tested the effect of different dosages of HS PBs in mangrove (*Avicennia germinans*) seedlings grown in Cd-contaminated solutions [94]. They found that HSs induced changes in mangrove root architecture and anatomy and that the optimal dose to maximize root length and area at each experimental stage was 4 mM C. The same dose maximized root development even in Cd-stress conditions [94]. In fact, the stimulation of root growth generally occurs at low HS dosages, while high dosages can gradually inhibit root development [95]. Based on the results obtained, Pittarello et al. [94] suggested that HS PBs can be useful in phytoremediation programs since they can favor phytoextraction by increasing root length and root surface area and phytostabilization by improving both soil oxygenation and the growth of root involved in the metal accumulation (cortex and aerenchyma) [94].

In a two-year study, the phytoremediation capacity of giant reed (*Arundo donax*) towards substrates polluted by heavy metals (Pb and Zn) was evaluated [96]. Leonardite-derived HA was added to the growth medium. The authors found that the biostimulant increased shoot and root plant biomass, raised the N content in culms, and stimulated bacterial soil growth. Furthermore, biostimulated giant reed accumulated higher Zn amounts in culms. The authors concluded that HA could be a valuable tool to improve phytoremediation and reduce its costs [96]. Dobbss et al. [95] tested the effect of vermicompost HS on the alleviation of iron (Fe) toxicity to aroeira (*Schinus terebinthifolius*) seedlings. Fe 250 μM in the hydroponic growth medium caused leaf chlorosis and reduction in plant growth. On the other hand, HS application significantly stimulated root and leaf development. In samples grown with both Fe and HSs, the symptoms of HM toxicity were alleviated [95]. In addition, these plants accumulated lower Fe and showed reduced antioxidant activities of the enzymes POD, CAT, and APX compared with samples treated with Fe or HSs alone. The authors concluded that HSs helped plants to prevent excessive Fe accumulation and that this material could be helpful in the recovery of HM-contaminated environments [95].

Evangelou et al. (2004) proposed the use of HA as an alternative to synthetic chelators to increase the solubility of metal cations in soil and their absorption by plants. In fact, synthetic chelators such as EDTA could have some negative effects that limit their use in phytoremediation: for example, they may have a toxic effect on plants, be non-selective in extracting metals or not be biodegradable [97]. The authors investigated the effect of increasing HA amounts in enhancing the phytoextraction capacity of tobacco (*Nicotiana tabacum*) plants in Cd-contaminated soils. Tobacco shoots biostimulated with the highest HA dosage significantly increased Cd accumulation. The authors concluded that HA could represent a viable alternative to synthetic chelators and that combining a natural chelator and suitable plant species (e.g., hyperaccumulators) can strongly accelerate the phytoextraction of pollutants and raise its efficiency [97].

Sung et al. [98] found that the application of humic acid increased the phytoremediation performance of *Phragmites communis* in wetlands polluted simultaneously by heavy metals (Pb, Cd, Cu, Ni) and petroleum hydrocarbons. HA reduced biomass losses due to the contaminants and significantly increased soil microbial activity. Furthermore, HA increased the metals’ bioavailability and their absorption by plants. The bioconcentration factor (BCF), estimated for all the metals investigated in this study, was significantly higher in both shoots and roots of biostimulated *P. communis* than in untreated samples. Moreover, HA strongly increased the total petroleum hydrocarbon (TPH) degradation in *P. communis*-planted soil. The authors concluded that this PB could be used to improve *P. communis* performance in phytoremediation and that the combination of HA with *P. communis* could be suitable for preventing groundwater contamination and protecting surrounding environments [98].

Bandiera et al. [99] tested the effect of two different concentrations of HA on fodder radish (*Raphanus sativus*) grown on HM-polluted (Co, Cu, Pb, Zn, and As) pyrite cinders. In the experiments, different methods of HA application were tested. As found in other studies [94], low amounts of HA positively affected plant growth and mitigated HM toxicity, while higher amounts provoked phytotoxic effects in plants. The authors hypothesized that higher dosages of HA increased the HM bioavailability in the growth medium. In accordance with this, biostimulated radish showed higher HM uptake and translocation to the aboveground tissues (especially Cu and Pb) and increased root elongation. Among the methods employed, the foliar HA application was the most effective in attenuating HM toxicity to plants and favoring their removal. Following these results, the authors suggested the potential of using HA in phytoremediation programs [99].

Moreno et al. [100] investigated Hg accumulation in plant species grown in Hg-polluted mine tailings. The application of growing concentrations of HA improved Hg solubility in the growth medium, especially when sulfur (S)-containing ligands (ammonium and sodium salts) were also added. The authors stated that this effect was probably due to the formation of Hg–thiosulphate and Hg–HA complexes [100]. Furthermore, Hg concentration in Indian mustard (*Brassica juncea*) roots increased following HA application significantly while root-to-shoot translocation was inhibited. The authors assumed that Hg–thiosulphate complexes were favored in translocation of the shoots, while Hg–HA complexes were retained in root tissues [100].

### 3.2. Protein and Amino Acid Hydrolysate-Derived PBs

Biostimulants based on protein hydrolysates (PHs) consist of a mixture of amino acids, peptides, polypeptides, and denatured proteins deriving from enzymatic, chemical, or thermal hydrolysis of animal- or plant-derived raw materials [70,73]. Source materials of PHs are primarily agro-industrial waste or by-products (e.g., crop residues or collagen); therefore, the production of PHs represents an attractive opportunity to valorize certain waste materials [70,101]. PH-based PBs mainly contain the amino acids alanine, arginine, glycine, proline, glutamate, glutamine, valine, and leucine. However, they can also include non-protein components, such as fats, carbohydrates, or macro- and micronutrients, which show biostimulatory actions [101]. PHs are currently used in agriculture since they (1) have a positive impact on soil microbial and enzymatic activities; (2) improve the mobility and solubility of microelements in soils; (3) enhance plant nutrient uptake and use efficiency as a consequence of the previous point; (4) stimulate carbon and nitrogen metabolism in plants; (5) enhance plant biomass production, with particular regard to the roots; (6) improve crop productivity [68,101]. In a very recent study, Rouphael et al. [102] compared the effect of PHs obtained from animal (A-PH) and plant (V-PH) sources at three equivalent nitrogen rates, finding a much more significant benefit on the growth of basil (*Ocimum basilicum*) plants treated with V-PH biostimulants. In fact, increased fresh weight, CO_2_ assimilation, and water use efficiency, and higher uptake and translocation of K, Mg, and S were found in V-PH-treated basil. On the other hand, decreased photosynthetic activity, plant growth, and biomass production were observed in A-PH-treated plants [102]. The potentially detrimental effect of some A-PH biostimulants, especially at high dosages, has been attributed to their high concentration of free amino acids, unbalanced amino acid composition, and high salinity content [101].

In conclusion, it can be affirmed that the ability of PHs to induce plant resistance and tolerance to various kinds of stress is now well known. The beneficial effects prompted by PHs also include reductions in HM toxicity to plants [101,103]. Despite this, no scientific study has to date demonstrated the effectiveness of these biostimulants in increasing the plant phytoremediation performance. Therefore, this field has yet to be studied. Furthermore, many things about PHs still need to be clarified, such as their mechanism of action which is not completely understood [104].

### 3.3. Inorganic Salt-Derived PBs

Among the non-biological derived biostimulants, several researchers are exploiting the potential of those derived from inorganic salts. One of the most studied is phosphite (Phi), an isostere of the phosphate anion (H_2_PO_4_^−^), in which one of the oxygen atoms bonded to the P atom is replaced by hydrogen [105]. Phi has been proved to improve nutrient uptake and assimilation and abiotic stress tolerance and promotes root growth. In addition, it is largely used for controlling pathogens [106].

The efficiency of NPK is also explored. *Pteridium aquilinum* (bracken fern) was tested for uptake of Cu from polluted water; at the end of the trial, several chemical parameters of water and the mortality of the fish *Clarias gariepinus* were assessed. With respect to control, NPK treatment improved water quality, even if a certain fish mortality rate was still observed [107]. NPK was also used to enhance the phytoremediation in *Spartina* sp. in sites polluted by crude oil [108]. Previously, several authors have found an increase in the rate of oil degradation, a possible result of increased microbial activity (as there is more availability of nutrients) and an increase in transplant biomass, both due to fertilizer addition. However, Ndimele and colleagues [108] did not find the same results, as NPK amendment did not show a significant effect on the phytoremediation. Regarding Cd phytoextraction, NPK fertilization was proved to spike the efficiency of *Sedum spectabile* to accumulate it in its aboveground tissue [109] and to enhance its uptake in *Cosmos sulphureus* and *Cosmos bipinnata* [110]. Similar results were observed in *Solanum nigrum*: the phytoextraction efficiency of the plant was significantly improved; furthermore, translocation was significantly enhanced in aboveground tissues compared to roots. *Solanum nigrum* could thus achieve higher phytoremediation abilities and Cd tolerance with the addition of NPK fertilizer [111]. It should be stressed that the application of chemical fertilizer with inappropriate composition has many limitations, i.e., improper mobility and availability of Cd in the plant–soil system, causing eutrophication in aquatic ecosystems and acidification of indigenous soil systems [112]. At the same time, the excessive uptake of Cd can also interfere with plant cellular metabolism (i.e., ROS production, inhibition of essential biomolecular functional groups), so the choice of the right fertilizer and the best hyperaccumulator is a crucial step.

Sulfur is a crucial element for plant growth and is deeply supplemented in agriculture; however, the excessive use of fertilizers to supply S can generate an ionic imbalance altering the soil, causing a nutritional deficit and contaminating aquifers by leaching [113]. Sodium thiosulfate (ST) can be a biostimulant that helps the plant prevent S deficit and, at the same time, could improve heavy metal tolerance and be useful to enhance phytoremediation of polluted soils. Navarro-Leon and colleagues used ST to enhance Cd accumulation in *Brassica* plants [113]: they showed that ST should not be used as a biostimulant because it reduced plant biomass, but it could be used for Cd phytoremediation purposes. Its good effect is dose-dependent, as the higher dose (4 mM) might saturate the transport systems that transport Cd to the shoot, and doses of 2 mM could enhance the phytoremediation efficiency. Phosphate (P) and thiosulfate were also studied for arsenic accumulation in the species *Brassica juncea* [114]. ST emerged as a good tool to improve As uptake, while P did not show interesting significant differences.

Sodium (ST) and ammonium thiosulfate (AT) were also studied in relation to Hg uptake in *Oxalis corniculata*: the first one halved the phytoremediation time, while AT reduced it by about 25% [115]. Even chlorides are inorganic salts that can be used as biostimulants, and clarifying their interaction with heavy metals is essential for controlling pollution and growing ‘‘metal-clean” foodstuffs. As an example, the presence of NaCl in the soil can modify the rhizosphere composition and the ability of the plant to uptake metals such as Cd. This has been recently proven in radish, where NaCl helped Cd^2+^ uptake in the hypocotyls [116]. NaCl was also demonstrated to be efficient in Cd^2+^ removal in *Conocarpus* [117]: the authors found that Cd^2+^ concentration increased both in shoot and root in the presence of NaCl. However, its translocation from root to shoot was not increased, rendering this tree suitable for the phytostabilization of Cd^2+^-contaminated saline soils [117].

Another study on *Chenopodium quinoa* showed that the plant exhibited improved growth and tolerance against Cd when grown at a salinity level of 150 mM NaCl. Salt relieved the quinoa plants from Cd-induced toxicity by inhibiting the aggregation of Cd and activation of the antioxidant enzyme system of the plants. Increased tolerance and less uptake of Cd due to moderate salinity levels showed that the quinoa genotype named ‘‘Puno’’ was suitable for the phytostabilization and could be successfully cultivated in Cd-contaminated saline soils. In contrast, an elevated concentration of salinity (300 mM NaCl), combined with Cd pollution, reduced shoot and root growth by more than 50%, caused overproduction of H_2_O_2_, and triggered lipid peroxidation [118]. Soil amendments such as limestone, dolomite, and chalcedonite can have a significant impact on the aided phytostabilization in acidic soils; this is the case for *Festuca rubra* and chromium (Cr), which is highly carcinogenic and thus crucial to remove [119]. These amendments, especially chalcedonite, have a good potential practical application because of their effectiveness in Cr immobilization; moreover, they help to recreate vegetation in degraded areas, as they were demonstrated to stimulate *Sinapis alba* germination and root growth.

### 3.4. Microbial-Derived PBS

**Beneficial bacteria**. The utilization of beneficial bacteria has been foreseen and implemented over the years. Bacterial roles in plant interactions are well known and exploited [120,121]; like fungi, bacteria can represent a continuum between mutualism and parasitism.

In agriculture, their use as biostimulants considers mainly two types of interactions, i.e., endosymbionts, such as *Rhizobium* and related taxa, and mutualistic plant growth-promoting rhizobacteria (PGPRs), also indicated as plant growth-promoting bacteria (PGPBs) that can also become endophytes. Rhizobia and related species are widely commercialized as biofertilizers, since they can fix nitrogen, facilitating nutrient acquisition by plants. In the scientific literature, as well as in the textbooks, the biology, molecular biology, and biochemistry of these microorganisms are extensively explained. PGPBs are considered multifunctional microorganisms, influencing several aspects of plant life, such as nutrition and growth, morphogenesis and development, response to biotic and abiotic stress, and interactions with other organisms in agroecosystems [121,122,123].

Bioremediation foresees the use of microorganisms for their ability to degrade environmental pollutants through their biochemical pathways related to the organisms’ activity and growth. PGPBs can positively interfere with HM uptake; in fact, the presence of PGPBs can also enhance abiotic stress tolerance [121,124] by alleviating metal-induced phytotoxicity, thus enhancing the biomass of plants grown in heavy metal-contaminated soils [125,126,127]. Therefore, PGPB-enhanced phytoremediation is considered a promising technology for remediating metal-polluted soils. We can screen bacterial strains that could adapt to the local environment and immobilize heavy metals. Some species of PGPBs (*Pseudomonas*, *Delftia*, *Enterobacter*, *Arthrobacter*, *Bacillus*) have shown high resistibility to Cd, and at the same time, they can decrease Cd bioaccumulation in plants by precipitating or absorbing Cd or enhancing root development [128,129,130,131,132,133,134]. However, there is still a need to screen and isolate newer PGPB strains able to immobilize Cd since the selected strains may not be able to perform well in different contaminated sites.

Even though bacterial species share useful bioremediation traits, a major limitation of their bioremediation efficiency may depend on factors that do not support the rapid growth of such beneficial bacterial populations, i.e., nutritional deficiency and competition by other bacteria. Laboratory investigation has been conducted in order to implement nutrient supply to microorganisms used to reduce HM contamination, and to provide optimal environmental conditions to these strains [135].

Recent research has also investigated the use of biochar as a carrier material for microbial inoculants, which can promote early colonization of the rhizosphere with beneficial microorganisms to overcome the bacterial growth constraints [136] and references therein). Under this perspective, some authors suggested this approach to help both bacterial growth and bioremediation activity [128,137]. The use of bacterial strains and biochar was utilized by Ma and collaborators [137]. In their study, a PGPB strain of *Bacillus* sp. TZ5, selected for Cd-immobilizing potential, was loaded on biochar; pot experiments with ryegrass indicated that the percentage of acetic acid-extractable Cd in biochar treatments significantly decreased by 11.34% with respect to the control.

In situ immobilization of Cd has been achieved by the use of a strain of *Pseudomonas chenduensis* and biochar; the supplementation of these two additives to paddy soil reduced the exchangeable/acid soluble Cd fraction and significantly decreased Cd availability; a reduction in the disturbance of soil microbial community under cadmium contamination was also observed [128].

The synthesis of biofilms in a single PGPB or consortia of PGPBs has been investigated and the reported ability to ameliorate plant drought tolerance might be effectively utilized in projects that foresee strategies for water conservation of plants [138]. The biofilms are composed of high molecular weight organic macromolecules, consisting mainly of exopolysaccharides (EPSs) with smaller proportions of protein and uronic acids. EPSs act as a protective barrier of bacteria towards environmental stresses, such as salinity, drought, heavy metal toxicity, etc. Several bacterial genera have been reported to produce EPS, among them *Agrobacterium* spp., *Xanthomonas campestris*, *Bacillus* spp., *Arthrobacter*, *seudomonas* spp., etc. [139,140]. EPSs may represent a powerful tool for the cleanup of toxic metals because of the presence of anionic groups characterized by metal ion chelation capabilities. Efficient HM remediation through bacterial EPS is based on the presence of non-neutral, negatively charged EPS (i.e., EPS with abundant anionic functional groups) [141]. The production of negatively charged EPS has been reported in different species [141].

Bacterial siderophores (Fe-complexing molecules) can enhance the mobility and reduce the bioavailability of HM with subsequent removal from the soil [142].

Besides the useful characteristics reported above, genetic engineering of bacteria has been applied in order to remove those heavy metals, such as Hg, that are not taken up by the bacteria [140,142].

**Cyanobacteria.** Several species of cyanobacteria have shown the ability to promote plant growth and ameliorate plant tolerance to abiotic and biotic stresses [143]. In the interaction with plants, cyanobacteria play a pivotal role in plant growth promotion by increasing the supply of different nutrients, including the fundamental trait of nitrogen fixation of some species (e.g., *Nostoc* sp. and *Anabaena* sp.), and the release of phytohormones. They can also enhance the water availability of the upper soil layer, thus improving its physicochemical conditions, as well as the release of EPSs that facilitate aggregation of soil particles and the accumulation of organic content. All together, these features ascribe to cyanobacteria a pivotal role in sustainable agriculture, which ranges from their use as biofertilizer to soil amendments for the recovery of infertile soils ([143] and references therein) [144]. Cyanobacteria’s bioremediation ability in waste waters and soils has also been reported. The pollutants removed successfully are heavy metals and pesticides [145]. The ability to remove HMs, such as Cr, Cu, Pb, and Zn, from coal fly ash has been described for the species *Anabaena variabilis*, *Nostoc muscorum*, *Aulosira fertilissima*, and *Tolypothrix tenuis* [146].

Similarly to bacterial EPSs, the cyanobacterial EPSs play a significant role in soil aggregation due to their gluing properties [147] and in binding ability for HMs [148] and sodium ions [149] that can improve plant development in saline or polluted soils. Seifikalhor et al. [150] reported that the application of *Spirulina platensis*, as corn seed priming treatment, improved plant growth, reduced Cd translocation from root to shoot, ameliorated photosynthetic electron flows, and increased non-photochemical quenching in Cd-exposed plants, thus mitigating the toxic effects on plants. The reduction in root Cd content of seed-coated plants was more than 90% 12 days after sowing.

Faisal and collaborators [151] suggested that the removal of Cr by *Oscillatoria* sp. and *Synechocystis* sp. is possibly involved in the observed increased wheat growth.

Regarding insecticide removal, cyanobacterial species, such as *Synechocystis* sp. and *Phormidium* sp., are capable of bioabsorbing and removing the systemic insecticide imidacloprid from the soil [152], while *Scytonema hofmanni* and *Fischerella* sp. can remove the insecticide methyl parathion [153,154].

On the other hand, many cyanobacterial genera have been studied for their toxin synthesis, which can represent a risk for human health, even though some cyanotoxins showed anticancer potential in human cell lines, providing interesting and promising results for future research, especially concerning the control of human adenocarcinomas [155]. Therefore, the use of cyanobacterial species needs a preliminary careful screening of the strains before being used in bioremediation.

### 3.5. Seaweed-Derived PBS

Algae can grow both autotrophically and heterotrophically and have large surface area/volume ratios, phototaxy, phytochelatin expression, and the potential for genetic manipulation. Based on these characteristics, algae are considered good candidates for biomonitoring and phytoremediation of polluted waters [156]. In addition, algae are able to remove and concentrate HMs since their large biomass production gives them a high sorption capacity [157]. Statistical analysis of algal biosorption reported potentiality absorption of about 15.3–84.6% by the algae, which is higher if compared to other microbial biosorbents [158]. Among the taxa, Phaephyceae are known to have high absorption capacity, being able to absorb metals such as Cd, Ni, and Pb through chemical groups present on their surface, such as carboxyl, sulfonate, amino, as well as sulfhydryl. Biosorption of metal ions occurs on the cell surface by means of ion exchange ability [158].

Moreover, algae are a source of polymers characterized by the presence of biologically active components, acting as agricultural biostimulants, which can be involved in the management of abiotic and biotic stresses in plants [159]. The bioactive compounds present in seaweed extracts (SEs) are beneficial to plants by promoting root and seedling growth in crops, and enhancing flowering and fruit production [160,161]. Being widely utilized as PBs, the seaweed extracts (SEs) represent more than 33% of the biostimulant global market [162]. Numerous taxa have been considered as potential platforms for biostimulant production and, in particular, beside red and green algal species, the kelp *Ascophyllum*, *Fucus*, and *Laminaria* are the dominant taxa [162].

Generally, PBs of algal origin are composed of polysaccharides extracted from different seaweeds species (e.g., *Ascophyllum nodosum*, *Ecklonia maxima*, *Durvillaea potatorum*, *Durvillaea antarctica*, *Fucus serratus*, *Himanthalia elongata*, *Laminaria digitata*, *L. hyperborea*, *Macrocystis pyrifera*, and *Sargassum* spp. ([159] and references therein). Depending on processing methods, SE may also contain minerals, phytohormones, vitamins, phenolic compounds, and antimicrobial agents [163,164]. Such diversity in the composition provides SE, either applied as foliar spray or on soil, with unique features that act positively on soil retention and remediation and as a source of nutrients. Moreover, some authors have evidenced the effect of SE on the down- and up-regulation of some key genes involved in response to abiotic stress, such as ROS scavenging-related genes, Na^+^ transporter and antiporters genes, and the hormone abscisic acid (ABA) [162].

The activities reported above confer to the SE-treated plants the possibility to improve biotic as well as abiotic stress response, the latter being fundamental in phytoremediation. Thus, the application of algal biostimulants on crops growing in HM-polluted soil may also positively affect, to some extent, the potential of heavy metal accumulation characteristics of certain crop species, even though contradictory reports are present in the literature [165].

Although few studies have been carried out concerning the use of algal PBs in phytoremediation, in a recent study a commercial seaweed-derived biostimulant (Megafol) was applied to duckweed (*Lemna minor*), a free-floating aquatic species, to increase the plant’s capacity to tolerate and remove the herbicide terbuthylazine (TBA) from polluted water [24]. This biostimulant derives from *Ascophyllum nodosum* with the addition of the amino acids proline and tryptophan, sugars, vitamins, and betaines [166]. Previous studies shed light on its ability to improve the plant resistance to various abiotic stresses [74,76] and, in particular, to herbicides [83]. In the cited study [24], the treatment of duckweed with the herbicide alone reduced plant proliferation and biomass production. On the contrary, biostimulated plants were less affected by the herbicide, thanks to the induction of some antioxidant enzymes (APX and CAT). Finally, phytofiltration experiments highlighted that the biostimulated duckweed removed higher amounts of TBA from polluted water with respect to the non-biostimulated plants treated with the herbicide. Based on these results, the authors concluded that Megafol successfully improved the duckweed phytoremediation potential, through the induction of defensive molecules [24].

Hu et al. [165] tested the effect of different concentrations of a commercial algal biostimulant on the Cd uptake of the accumulator *Nasturtium officinale*. Besides plant growth-promoting activity and the enhancement of photosynthetic pigments at all PB dosages, the biostimulant increased Cd extraction by plant roots. In *N*. *officinale* shoots, Cd content decreased for the inhibited translocation of Cd to the shoots. Therefore, the authors concluded that this PB might not be suitable for enhancing the phytoremediation ability of *N. officinale* towards Cd-contaminated soils. Nevertheless, the results indicated that this biostimulant might be used to cultivate vegetables in Cd-contaminated soil [165].

### 3.6. Plant Extract-Derived PBs

Among the compounds of plant origins that may act as biostimulants and ameliorate the phytoremediation activity, a clear example is provided by melatonin (N-acetyl-5-methoxytryptamine), a ubiquitous molecule presents in prokaryotes and eukaryotes. In plants, the important roles of melatonin are related to both antioxidant activity and redox network regulation. Acting as a biostimulator, especially under biotic or abiotic stress conditions, exogenous melatonin application to plants can improve the uptake of phosphorus, nitrogen, and sulfur, and at the same time, minimize the harmful effects of the stressors, by controlling the levels of reactive oxygen species (ROS) through the activation of antioxidant response and by mobilizing toxic metals through phytochelatins [167].

Activating agents can be successfully applied to improve phytoremediation activity. Li et al. [168] described an improved efficiency of extraction technology by *Sedum alfredii* in experiments, where the effects of two plant extracts (i.e., *Oxalis corniculata*, OX, and *Medicago sativa* extract, ME) and citric acid were tested. The application of these three activating agents was beneficial for the decontamination of Cd and Zn in soils, showing an improved repairing efficiency by 3.92, 3.37, 3.33 times and 0.44, 0.20, 0.86 times, respectively. Moreover, OX and ME did not have harmful effects on soil properties and plants, since they did not alter chlorophyll fluorescence parameters, while CA improved F0, but significantly reduced Fv/Fm. According to these authors, the combination of plant extracts and hyperaccumulators can more efficiently remove heavy metals from contaminated soils and provide a further tool for mitigation of soil pollution.

A possible perspective for future application will be the utilization of the C4 species *Miscanthus* × *giganteus* besides its non-food use (i.e., biofuel, the pulp of cellulose [169]) as considered by Técher et al. [170]. In their study, the effects of root exudates of *Miscanthus* on biostimulation of PAH degradation was tested. Four bacterial consortia with different co-metabolic degradation abilities were characterized and tested for exudate biostimulation. The authors measured bacterial growth and relative degradation activity (through the production of intermediate metabolites) in the presence of PAH and plant secretions, and the tests were carried out in a specifically designed microplate assay. The analysis of the polyphenolic components of exudates indicated the presence of a diverse range of flavonoid-derived compounds. Among them, two identified molecules, quercetin and rutin, played a major role in promoting bacterial growth and PAH metabolism.

Besides increasing crop yield and ameliorating plant tolerance to stress, PBs can also affect the microbiota, which, as reported above, plays a crucial role in plants’ fitness. In a study, Luziatelli and coworkers [171] evaluated the effect of commercial products (i.e., a vegetal-derived protein hydrolysate (PH), a vegetal-derived PH enriched with copper (Cu-PH), and a tropical plant extract enriched with micronutrients (PE)) on *Lactuca sativa* plant growth and the ability of these products to enhance the growth of beneficial or harmful bacteria. Based on the enhancement of shoot biomass of lettuce, the results confirmed the biostimulating effect of the three products. The foliar application of the products stimulated the growth of specific bacteria belonging to *Pantoea*, *Pseudomonas*, *Acinetobacter*, and *Bacillus* genera, thus altering the composition of the microbial population. Some of the identified strains possessed PGP characteristics, therefore, the findings of the study indicated that the commercial organic-based products could enhance the growth of beneficial bacteria occurring in the plant microbiota, while no harmful bacterial strains were detected. Based on these results, further studies should be undertaken to better foresee the effects on other microbiota by different PBs.

### 3.7. Fungal PBS

As previously described, plant biostimulants are formulated with diverse microorganisms and/or substances that are applied to crops; among fungal species, *Trichoderma*-based products have been particularly successful based on biostimulating activity and the capacity to control phytopathogenic fungi and ameliorate the tolerance to abiotic stresses [172]. Considered safe for humans, livestock, and crop plants, both solid and liquid formulations containing conidia can be used to produce suitable quantities of active and viable inocula for product formulation and field use. Biostimulant properties of *Trichoderma* depend on fungus–root communication via volatiles, ethylene, and auxins. Proteomic and genetic data suggest that *Trichoderma* activates different enzymes, DNA processing proteins, and transcription factors in plants.

Among the mutualistic associations, mycorrhizal fungi, to whom belong a heterogeneous group of taxa, establish symbiosis with over 90% of plant species. In particular, the arbuscule-forming mycorrhiza (AMF), a type of endomycorrhiza associated with the majority of crop plants, can act as biofertilizers by absorbing and translocating mineral nutrients to plants and induce changes in secondary metabolism leading to improved nutraceutical compounds. Additionally, by interfering with the phytohormone balance of the host, AMF may influence plant development (bioregulators), thus inducing tolerance to soil and environmental stresses (bioprotector) [80]. Beyond bioprotectant activity, it is noteworthy to add the role played by AMF in decreasing the detrimental effect of pollutants, such as heavy metals. In fact, the contaminants can be immobilized in fungal biomass, providing a further benefit to plants and introducing the possibility of utilizing mychorrized plants in phytoremediation [173].

Fungal compost may also play a positive role in phytoremediation. Spent compost (spent mushroom compost, SMC) of *Pleurotus ostreatus* was tested by Asemoloye et al. [174,175]. The effect of SMC on phytoremediation potential was determined in *Megathyrsus maximus* Jacq. (Guinea grass) grown in heavy metal- and PAH-polluted soils. The effect of SMC (0, 10, 20, 30, and 40%) treatments on chemical characteristics of the soil was determined through soil analysis before and after the experiment. The results suggested that SMC treatment modified soil chemical characteristics and improved plant growth, biomass production, and phytoremediation potential to different degrees concerning the amount of SMC applied to the soil. The positive action of SMC as organic compost may be based on the enhancement of the metal solubility and/or uptake by plants, through either metal chelator activity or stimulation of microbial activity in the rhizosphere. Moreover, the biostimulatory activity should also positively promote the co-degradation of hydrocarbon [174]. The authors suggested the utilization of SMC for soil stimulation and the improvement of phytoremediation.

## 4. Conclusions

The preservation of natural resources is a priority that can no longer be postponed due to their worrying state of degradation. The constant release and subsequent accumulation of toxic substances in the environment must undoubtedly be counted among the main causes of environmental degradation and deterioration of natural resources. Given the importance of these resources, one of the most relevant actions to be developed more carefully and then implemented concerns environmental remediation. To this end, new technologies are needed that are environmentally friendly and do not impact the environment. Furthermore, they should allow the effective removal and cleaning of sites polluted by contaminants.

Among green remediation technologies, phytoremediation has gained particular importance due to its economic sustainability and environmental friendliness. However, this technology still has some weaknesses, as it can be slow and ineffective in completely removing contaminants. Therefore, in this review, we have proposed biostimulants among the emerging tools that can improve phytoremediation and make it more effective. In fact, since these compounds are used to mitigate the effect of many different toxic substances on plants, the scientific literature shows that they can enhance the plant’s ability to remediate contaminated environments. The wide range of sources from which biostimulants can be obtained can offer the prospect of testing many PBs with different modes of action. In the present work, studies that have tested the effectiveness of PBs derived from humic substances, protein and amino acid hydrolysates, inorganic salts, microbes, seaweeds, plant extracts, and fungi in phytoremediation are reviewed and discussed. These studies highlight how biostimulants can, in some cases, promote pollutant uptake and increase the removal process, while helping the plant overcome the stress resulting from the presence of the xenobiotic. In other cases, PBs can limit the uptake of the contaminant, thus allowing the plant to better survive the resulting damage. In most of the cases analyzed, the choice of the right dosage of PB is critical to the success of the phytodepuration process. In some studies that considered environmental pollution from organic substances, the stimulatory action of PBs on pollutant biodegradation has been reported. This action is also due to the positive effect of PBs on the microbiota. Not all types of PBs have yet been tested in phytoremediation; further studies are needed to expand our knowledge in this field.

The many advantages of using biostimulants in phytoremediation have been extensively explained above. Summing them up briefly, these substances are able to reduce the toxicity of certain compounds in the plant, including by activating antioxidant-like responses. Moreover, the application of these compounds on the plant or on its growth medium is extremely easy, even on a full scale and in a real phytoremediation system, falling among the common agricultural practices. Finally, given the totally natural origin of PBs, their use does not adversely affect the environment, and it does not require their recovery. The only disadvantage lies in the additional costs, related to the purchase of PBs and to their application. However, in our opinion, which is based on the literature studies analyzed and reported in this review, this effort is definitely rewarded by the aforementioned positive effects.

**Future perspectives.** It is worth mentioning the possibility of obtaining bioactive substances from agroindustrial waste or by-products. This last challenge responds to the need to move towards a circular economy that allows the valorization of materials that otherwise would have to be disposed of, creating a further significant pressure on the environment. In addition, future research should be directed toward investigating the mechanisms of action that allow biostimulants to carry out a better cleaning of polluted environments.

Although studies reported here have shown that biostimulants can effectively improve the phytoremediation potential of some species or improve their tolerance to the toxicants, this is still an open field where substantial research work needs to be carried out to understand how the use of these materials could be optimized for a successful application in the field. In addition to laboratory experiments, the scaling up of phytoremediation systems using biostimulants is needed and would allow the determination of the real effectiveness of the systems proposed in this review.

We believe that the studies mentioned above are necessary to increase the knowledge in the area of phytoremediation assisted by PBs, and consequently to enable their real use in polluted environment cleanup practices.

## Figures and Tables

**Figure 1 plants-11-01946-f001:**
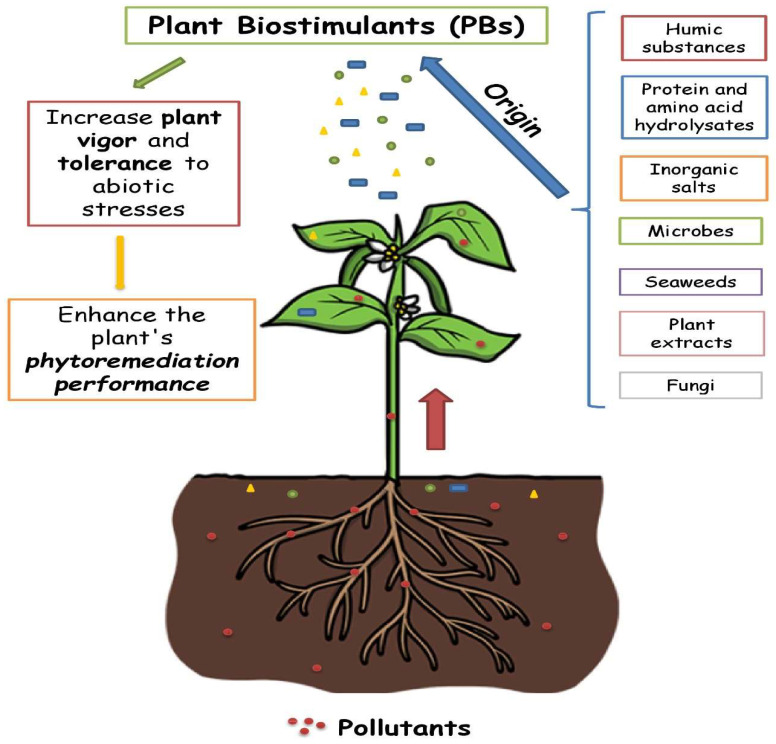
Beneficial effects of different origin PBs on plant response to stress and phytoremediation activity.

**Table 1 plants-11-01946-t001:** Advantages and disadvantages of phytoremediation.

Advantages	Disadvantages
Suitable for various types of contaminants (organic substances, metals, metalloids, dyes, hydrocarbons, radioactive substances)	Not applicable in some circumstances (for example, when contaminants are found in deep soil layers, not accessible to the roots)
Efficient	Contaminants cannot be completely removed
Relatively cheap	Slower than conventional methods
Environmentally friendly	Not convenient for heavily polluted sites (due to the limited tolerance of a plant to pollutants)
Non-destructive	Difficult to apply when the pollutant is not completely bioavailable
Non-invasive	Strictly dependent on the environmental conditions
Aesthetically pleasing	The handling and disposal of harvested plant tissues could be problematic
Directly applicable in situ	Still under development (its potential has not been fully exploited)
Does not require energy	Commercial-scale applications of this technology are few and still inadequate
Can be used to remove more than one pollutant at the same time	
Has minimal equipment requirements	
Can be combined with other methods, such as conventional technologies	
Contaminants can be recovered from the plant tissues and marketed	
Provides habitats for animals	
Stimulates beneficial microbes	
Reduces soil erosion, simultaneously improving its structure and fertility	
Contributes to carbon sequestration	

**Table 2 plants-11-01946-t002:** Effects of PBs in ameliorating the stress response generated by pollutants in plants.

Plant Species	PB	Pollutant	PB Recommended Dose	Results	Ref.
Maize	Humic substances	Cr	4 mM C HA L^−1^	- CAT and proline increases - higher transcription of genes associated with stress signaling and response - higher biomass production	[79]
Maize	Silymarin-based biostimulant	Cd	0.24 g L^−1^	- increased photosynthesis efficiency - restored hormonal homeostasis - increased activities of antioxidants and enzyme gene expression	[82]
Maize	Megafol	Metolachlor	2.5 L ha^−1^	- lower levels of lipid membrane peroxidation - increased germination, biomass production, and vigor index - induction of antioxidant enzymes (APX, GPX, CAT)	[83]
Soybean	Fertiacyl Pòs	Glyphosate	0.4 L ha^−1^	- limited yield losses - limited symptoms of chlorosis and necrosis	[84]
Sunflower	Protein hydrolysates	Imazamox	3 L ha^−1^	- restoring the net photosynthetic rate, stomatal conductance, chlorophyll content, and plant growth	[86]

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
