# Peer review of "Use of Biostimulants as a New Approach for the Improvement of Phytoremediation Performance—A Review"

_plants, 2022, doi:10.3390/plants11151946_

Round 1

Reviewer 1 Report

Review of the paper: “Use of biostimulants as a new approach in phytoremediation-A review”  submitted to  Plants by  Maria Luce Bartucca, Martina Cerri, Daniele Del Buono and Cinzia Formi

General comments: 

 Phytoremediation has made significant progress in recent years by clarifying a number of polluted situations. Phytoremediation technology has a number of advantages that allow you to find a niche for its best suitability, however, it has weaknesses, one of them is the length of the remediation process. Contractors are looking for any opportunities to improve its performance in order to shorten its implementation period. The study presented for review is a collection of achievements in research and practice on how to make the phytoremediation process more efficient and, consequently, make it shorter.

Comments addressed to parts of the paper:

Partial changes to the title of the manuscript should be considered, as many of the applications described are already in use, so they are not a new idea. I suppose terms such as intensifier or improvement are more appropriate to the content of the review.

In the first part dedicated to metals and metalloids, I miss the solution practiced in the field of arsenic remediation. To get to the plant, this metalloid uses channels through which phosphorus penetrates, due to the significant amounts of phosphorus taken up, it is not so precise. This insidious attitude can be used, however, and in a situation when we want to take more arsenic, we limit fertilization to the necessary minimum, and on the contrary, when we want to limit As in harvested plants intended for consumption, the fertilization of these plants must be optimal. In my opinion, this procedure should not be missing from the presented study

I would also like to emphasize that the idea of collecting biologicals is very good because it is based on processes taking place in nature and is a good contrast to the proposed industrial methods, which often involve the introduction of additional chemicals and treated by many people as additional pollutants

.

            Final opinion of the review: 

In my opinion, the submitted review of the use of biostimulants is a valuable move for further progress in the development of phytoremediation and in this form, the article after taking into account the posted comments is suitable for publication in Plants.

Author Response

Referee 1

  • Partial changes to the title of the manuscript should be considered, as many of the applications described are already in use, so they are not a new idea. I suppose terms such as intensifier or improvement are more appropriate to the content of the review.

R: The title has been changed following the reviewer’s suggestion

  • In the first part dedicated to metals and metalloids, I miss the solution practiced in the field of arsenic remediation. To get to the plant, this metalloid uses channels through which phosphorus penetrates, due to the significant amounts of phosphorus taken up, it is not so precise. This insidious attitude can be used, however, and in a situation when we want to take more arsenic, we limit fertilization to the necessary minimum, and on the contrary, when we want to limit As in harvested plants intended for consumption, the fertilization of these plants must be optimal. In my opinion, this procedure should not be missing from the presented study

R: As requested by reviewer 1, a paragraph on arsenic and phosphorus competition in plants and the resulting following applications in phytoremediation has been inserted in the part related to “phytoextraction” (Par. 1.2. Phytoremediation)

Reviewer 2 Report

The review topic is really interesting and the manuscript is well written. Therefore, the manuscript has some problems that are listed below:

1) The goals of the review are not clear in the abstract. Lines 15-20 are a little bit confusing. Please, rewrite this part to improve the abstract of your manuscript.

2) The authors can bring more of the highlights found in the review, the text provided (Lines 21-23) is not enough. What are the PBs more tested and for the remediation of which substance?

3) Please, put the advantages and disadvantages of phytoremediation (Lines 144-176) in a Table. It will facilitate the reading of your manuscript.

4) Please, add a table on the topic ‘2.1 PBs in helping crops cope with toxic compounds’ with examples of published studies. For example, a table containing plants/PB (concentration, time)/ pollutant/ results/ reference

5) Please, add the unit of the molecular weight (Lines 369-372).

6) The topic 3. PBs for phytoremediation should have also a Table or a Figure. It will facilitate the reading of your manuscript.

7) Cyanobacteria (Line 619) have cyanotoxin and it was not explained in the review. Are Cyanobacteria suitable to be used as PBs even with toxic compounds inside their cells?

8) Please, create a section with Future perspectives about the PBs application. What do the authors think about the next steps in this topic?

9) The authors should be more critical of this topic and discuss it in the manuscript. What are the advantages and disadvantages of the PBs application? Is suitable the application of protein hydrolysates in a real treatment system? This discussion will bring the real perspective of PBs application.

Author Response

  • The goals of the review are not clear in the abstract. Lines 15-20 are a little bit confusing. Please, rewrite this part to improve the abstract of your manuscript.

R: Lines 15-20 of the abstract have been rephrased and rewritten accordingly to the reviewer request

  • The authors can bring more of the highlights found in the review, the text provided (Lines 21-23) is not enough. What are the PBs more tested and for the remediation of which substance?

R: This part of the abstract has been completely rewritten. The highlights found in the review have been specified following the reviewer’s indication

  • Please, put the advantages and disadvantages of phytoremediation (Lines 144-176) in a Table. It will facilitate the reading of your manuscript.

R: A table reporting the advantages and disadvantages of phytoremediation has been inserted in the text, as requested by the reviewer.

  • Please, add a table on the topic ‘2.1 PBs in helping crops cope with toxic compounds’ with examples of published studies. For example, a table containing plants/PB (concentration, time)/ pollutant/ results/ reference

R: A table on the topic 2.1 has been inserted as requested by the reviewer.

  • Please, add the unit of the molecular weight (Lines 369-372).

R: m.w. have been added as requested by reviewer 2

  • The topic 3. PBs for phytoremediation should have also a Table or a Figure. It will facilitate the reading of your manuscript.

R: A figure on topic 3 has been added as requested by reviewer 2

  • Cyanobacteria (Line 619) have cyanotoxin and it was not explained in the review. Are Cyanobacteria suitable to be used as PBs even with toxic compounds inside their cells?

R: Not all the cyanobacteria produce toxins, a careful screening should be performed to determine the “safe” strains to be used in phytoremediation. A comment has been added.

  • Please, create a section with Future perspectives about the PBs application. What do the authors think about the next steps in this topic?

R: A section on future perspectives about the PBs application in phytoremediation and our thoughts on next steps to be carried out has been added

9) The authors should be more critical of this topic and discuss it in the manuscript. What are the advantages and disadvantages of the PBs application? Is suitable the application of protein hydrolysates in a real treatment system? This discussion will bring the real perspective of PBs application.

R: A paragraph relative to  the advantages and disadvantages of the PBs application, even in a real treatment system, has been added in the “Conclusion”

Round 2

Reviewer 2 Report

The authors improved significantly the manuscript quality